# Technical note: Water table mapping accounting for river-aquifer connectivity and human pressure

Maillot Mathias[1-2], Nicolas Flipo[1], Agnès Rivière[1], Nicolas Desassis[1], Didier Renard[1], Patrick Goblet[1], and Marc Vincent[2]

[1]Geosciences Department, MINES ParisTech, PSL University, Fontainebleau, France
[2]EPTB Seine Grands Lacs, Paris, France

**Correspondence:** Maillot Mathias (mathias.maillot@mines-paristech.fr), Nicolas Flipo (nicolas.flipo@mines-paristech.fr)

**Abstract.** A water table mapping method that accounts for surface water-groundwater (SW-GW) connectivity and human pressure, such as pumping and underground structures occurrence, has been elaborated and tested in the heavily urbanized Parisian area. The method developed here consists in two steps. First, hard data (hydraulic head) and soft data (dry wells) are used as conditioning points for the estimation of the SW-GW connection status. A disconnection criteria of 0.75 m is adjusted on observed unsaturated zone depth (UZD). It is a default value in areas where such data are missing. The second step consists in the final mapping of water table. Given the knowledge of the disconnection criteria, the final map is achieved with an ordinary kriging of the UZD that integrates the surface water elevation as a nil unsaturated zone where it is relevant. The methodology is demonstrated on two datasets of UZD observations that were collected under low and high flow conditions.

## 1 Introduction

Water table maps are key tools for water resources and flood risk management. A way to characterize a water table distribution is to describe it using piezometric maps. Albeit this seems an obvious statement, some methodological aspects require further development, such as the way how to take into account uncertainty about surface water (SW) and groundwater (GW) connectivity.

This connectivity status can be either connected~~or disconnected~~, transitional or disconnected (Dillon and Liggett, 1983; Fox and Durnfor For the connected case, the surface water elevation corresponds to the water table ~~below the riverbed~~ and should be accounted as an observation sample ~~(Chung and Rogers, 2012)~~(Chung and Rogers, 2012; Winter et al., 1998), whereas surface water level should not be considered into mapping in the disconnected case (Hentati et al., 2016).

The river-aquifer connectivity status depends on hydrological and geological parameters such as the surface water level, water table, riverbed geometry and hydrogeological parameters of the substratum (Brunner et al., 2009; Peterson and Wilson, 1988; Rivière et al., 2014). Water table and surface water level distribution results from precipitation, recharge of aquifers, topography, riverbed and aquifer geometries, and hydrodynamic parameters ~~(Flipo et al., 2014)~~(Flipo et al., 2014; Bresciani et al., 2016). Urban GW are seriously affected by the development of urban areas ~~. Indeed~~in several ways. Besides the barrier effect induced by the occurrence of underground structures across groundwater flow, some modification of water budget may be caused by the

interaction between groundwater and these underground structures (Morris et al., 2003; Attard et al., 2016). For instance, leaky sewer and water supply plumbing networks may act as recharge (Abderrahman, 2006) or drain (in the case of sewers), limiting the water table rising above the structure (Dassargues, 1997). Generally speaking, human settlement nearby fluvial environments results in significant SW and GW decline due to pumping wells for domestic and industrial usages, as well as for under-
ground structure protection and the construction of underground infrastructures (Morris et al., 2003; Attard et al., 2016; Machiwal et al., 20
Moreover, the development of ~~embankments~~ levees along the river and riverbed dredging generate major modifications of the stream-aquifer status. So far, all those aspects have not been taken into account in water table mapping methodologies.

The most commonly used methods for the estimation of a continuous variable are ~~usual~~ linear estimators, neural network and kriging (Varouchakis and Hristopulos, 2013). The main linear estimators are inverse distance weighting (Gambolati and Volpi 1979, Philip and Watson 1986, Rouhani 1986, Buchanan and Triantafilis 2009, Sun et al. 2009) and influence polygon or moving average (Vicente-Serrano et al., 2003). ~~Varouchakis and Hristopulos (2013) compared these different methodologies and showed that kriging provides better performance~~ These different methodologies were compared in several studies and kriging was found out as a better estimator in terms of cross-validation ~~than~~ and performance than the other linear interpolators (Varouchakis and Hristopulos, 2013; Emadi and Baghernejad, 2014; Adhikary and Dash, 2017; Ohmer et al., 2017) . Although the linear estimation methods provide unbiased results, they do not account for the spatial heterogeneity of the samples distribution. The estimated value depends either on the nearest sampled value (influence polygon), or on every sampled values surrounding the estimation point (moving average) regardless the distance between the estimation point and each individual sampling point. Inverse distance weighting involves the arbitrary choice of the distance degree. The distance degree is a conditioning setting for the variability of estimated fields whereas kriging involves a weighting of observation that is consistent with the spatial distribution of the variable.

Recently, interpolations based on fuzzy logic or neural network derived methods have been tested (Kurtulus and Flipo, 2012; Sun et al., 2009). These methods are still suffering of a main drawback, that is they produce results without coherent spatial error structures (Flipo and Kurtulus, 2011). The diffusion kernel interpolation method used in Bresciani et al. (2018) showed good results for large datasets. This method is based on geographically weighted regression which aims to map the trend of a variable (Gribov and Krivoruchko, 2011). Depending of the used parameter in the application of this methodology, the produced map can be very smoothed or noisy. This method allows for the spatial representation of estimation error, nevertheless there is no guarantee that the resulting map honors the input data.

A widely accepted solution that provides information on estimation errors is kriging (Chilès and Delfiner, 1999; Matheron, 1955). It can be applied on different types of variables (Cressie, 1990) including water table (Hoeksema et al., 1989). Many studies produced water table maps resulting from kriging in order to describe water table distributions (Ahmadi and Sedghamiz, 2007; Bhat et al., 2014; Buchanan and Triantafilis, 2009; Chung and Rogers, 2012; Hentati et al., 2016; Hoeksema et al., 1989; Kurtulus and Flipo, 2012; Mouhri et al., 2013; Zhang et al., 2018). Rouhani and Myers (1990) noticed that water table data displays spatial nonstationarities, which are due to the ~~topographic slope~~ directional trends in hydraulic head gradients. Such nonstationarities cause problems in the determination of the experimental variogram and also generate large standard deviations of the estimation errors. A way to overcome the issues linked to nonstationarities was proposed by Desbarats et al. (2002). Their

methodology also based on kriging was developed for an unconfined aquifer. It relies on the spatial correlation between the water table and the topographic surface (King, 1899; Toth, 1962). This assumption was established by Desbarats et al. (2002) at large scales considering several watersheds. Haitjema and Mitchell-Bruker (2005) proposed that: "shallow aquifers in flat or gently rolling terrain may exhibit a relatively low $R/k$ ratio and still exhibit a water table that seems a subdued replica of

the terrain surface"(Haitjema and Mitchell-Bruker, 2005, p786), where $R[m/d]$ is the areal recharge rate, and $k[m/d]$ is the aquifer hydraulic conductivity. This methodology, that targets the unsaturated zone depth (UZD) instead of the hydraulic head, leads to lower values of the standard deviation of the estimation error for unconfined aquifer in non-urbanized area (Kurtulus and Flipo, 2012; Mouhri et al., 2013; Rivest et al., 2008; Sağir and Kurtuluş, 2017).

In urbanized area, the pumping of GW implies the decline of water table, which could lead to the ~~dry~~ drying out of a few
piezometers. The knowledge of a dry well can be added to a dataset in the form of an inequality (i.e. UZD larger than the well depth) (Michalak, 2008). The counter part of accounting for such information translated into a mathematical inequality is that it is incompatible with kriging itself. Therefore another methodology has to be used for water table mapping in such environments.

A solution is the usage of multiple conditional simulations that provides a conditional expectancy map of the variable. Its
application in hydrogeology was demonstrated for hydrofacies determination (Dagan, 1982), converting lithofacies into hydrofacies to constrain groundwater flow models. This study proved that the use of conditional probability reduces the variance of possible values of the targeted variable, for instance here hydrofacies properties. This methodology was applied in different geological contexts (Tsai and Li, 2007; Dafflon et al., 2008) proving its robustness and has not been applied to the UZD so far.

~~Another source of uncertainty in water table mapping methodology is the fact that over a large area, such as a watershed or~~
~~basin, the water table distribution is also driven by the recharge rate of the aquifer (Haitjema and Mitchell-Bruker, 2005). To~~ ~~avoid this drawback, our methodology assumes a nil recharge, which is the case in urbanized areas where a high degree of soil~~ ~~sealing is observed.~~

The mapping methodology presented in this paper relies on the assumption that the UZD variable is related to the topographic elevation and the river water level. ~~One~~ The second assumption is that UZD is not related to the stream water level in the case of
a disconnected hyporheic zone. Therefore, it can be applied to superficial aquifer units submitted to human pressures and other locations where the SW-GW connectivity is uncertain. The following questions are addressed: (i) which ~~method is the most~~ ~~relevant~~ methodological steps are required for water table mapping in alluvial plains? (ii) how to account for human practices such as pumping in the mapping methodology ? (iii) how to ~~define~~ determine the SW-GW connection status? (iv) finally, what are the consequences of such methodological refinements on produced maps of water table linked to hydrological events?

## 2  Mapping Methodology

Water table mapping was initially developed for the description of regional aquifers into natural or pristine environments. The usual way of mapping a water table is to use synchronous UZD measurements resulting from snapshot campaigns. The synchronization of measurements is crucial to avoid experimental bias (Tóth, 2002). This section describes a methodology

that combines conditional simulations of UZD, with an assessment of SW-GW connectivity and a final ordinary kriging of the UZD. Geostatistical processings are performed using the RGeostats R package (Renard et al., 2001 - 2019).

Fig. 1. describes the methodology. Firstly, the ~~dataset analysis is achieved in order to constitute the raw dataset for mapping~~raw dataset is composed of each measured UZD for the corresponding measurement campaign. The raw dataset is then transformed into a Gaussian score dataset using an anamorphosis function fitting in order to obtain a Gaussian probability density function (Chilès and Delfiner, 1999). Inequality constrained samples (dry wells) are estimated using a Gibbs sampling of the Gaussian score ~~dataset~~ subset (Geman and Geman, 1984; Freulon and de Fouquet, 1993). Thereafter, one hundred turning band simulations (Matheron, 1973) are performed and averaged before their backtransformation into the real data. A first guess map of water table is obtained averaging all back transformed simulations. The SW-GW connectivity status is deduced from the first guess map following a new connectivity criteria that permits to constitute the final UZD dataset. The final water table map is finally produced performing an ordinary kriging of the final UZD dataset~~that is removed~~. It is the step where UZD map is converted into water table map, subtracting the UZD from a reference Digital Elevation Model (DEM) of the ground.

## 2.1 First Guess - Simulations without considering the river water level

The initial dataset is made of hard data and soft data. The hard data are UZD measured during snapshot campaigns. The soft data are dry well depths. The dataset is characterized in terms of spatial statistics in order to justify the use of an appropriate geostatistical tool. UZD is defined in terms of a non-Gaussian probability density function conditioned with non-negativity constraint. ~~Unlike water table , UZD can be considered as a continuous stationary variable~~

Water table and UZD variables may show some directional non-stationarities at the local scale, especially looking in the same direction than the directional gradients. Nevertheless, the directional gradients in UZD are much less pronounced than the water table gradients, making it more amenable to treatment with stationary geostatistics. In the cases where a significant trend in the data is identified, the interpolation must be carried out using other geostatistical approach, such as universal kriging, that deals with the non-stationarity of the variable (Goovaerts, 1999).

### 2.1.1 Input data pre-processing & DEM smoothing

The use of UZD as a variable ~~to map~~ for mapping the water table requires to refer to the elevation of the ground from which water table can be ~~deduced from~~computed. In our approach the elevation of the ground is approximated using a smoothed DEM, called reference DEM. It is obtained merging a DEM and river water levels. This merged DEM is smoothed (Fig. 1., step 1) using SAGA GIS algorithm (Conrad et al., 2015) for moving average filtering~~(the search radius is~~, this methodology was already proposed by Mouhri et al. (2013). The smoothing of the DEM is required to avoid the occurrence of high frequency topography signals that would not be relevant with the ~~average width value of the stream network) .~~water table signal. The search radius is defined regarding two conditions: i) the DEM has to be smoothed enough to remove its high-resolution noise and ii) the information of river water level must be conserved in the final product. We tested several radii to fit these conditions and found out an appropriate value of 325 m.

The difference between rough DEM and smoothed DEM may be important in locations where the topographic slope is the most important. These locations include crucial areas nearby the riverbanks. Therefore, this difference is calculated at each sampling point. Due to the use of UZD, this generates a biased estimation of water table at these locations, given that this difference is not yet accounted for into the UZD measured value. The way to tackle the DEM smoothing effect is to constitute a first data subset, deducting the difference between smoothed DEM data and true wellhead elevation from the raw UZD data before proceeding with the next steps of our procedure (Fig. 1). For the sake of readability, this first data subset will still be called UZD raw dataset in the remaining of the paper.

### 2.1.2 Hard data selection & variograms

The variographic analysis of the UZD raw dataset is achieved in order to describe the variability of UZD in a 2D domain. In urbanized area, anthropic pressure such as permanent pumping, affects the natural correlation between DEM and UZD with the occurrence of local piezometric depletions. In terms of experimental variogram, the use of samples affected by anthropic pressure induces a drastic increase of the semi-variogram value. This cannot be considered as a representative variability of the UZD variable. To prevent this effect on the experimental variogram calculation, the original dataset is divided into two categories (Fig. 1., step 2). The first category regroups all samples where the UZD value is affected by the pumping wells. The second category is composed by the other samples. Information about the locations of pumping wells is required to identify these samples. In this study, the locations and pumping flow-rates are not available. The affected and unaffected piezometers are differentiated regarding the correlation between topography and water table. Grubb (1993) stated that water table within the capture zone of a pumping well is not hydrostatic, then it is assumed that topography and water table are not correlated within this capture zone. The samples where there is no correlation between topography and water table are identified as the affected samples. In this study, samples with a UZD value exceeding 10 m were found in that category. Note that this value may vary according to the case study. This differentiation is required to elaborate a geostatistical tool (i.e. variogram model) that only depends of natural variability. Therefore, all the variographic studies are performed on this second category called unaffected UZD dataset. While the above procedure was used to roughly approximate which wells are affected by pumping, any future applications of the method outlined in this technical note should identify the wells impacted by pumping using actual data on pumping rates and locations.

The experimental variograms are calculated on two types of variables: the Gaussian score used in the Gibbs sampling and conditional simulations, and the unaffected UZD dataset for the final ordinary kriging procedure. The Gaussian score variable used for Gibbs sampling-conditionnal simulation steps is described in the next subsections. UZD is the variable ultimately used for ordinary kriging. Each calculated experimental variogram is a representation of the spatial variability of the dataset. A variogram model is fitted to each experimental variogram with a composition of spherical, exponential and cubic functions. The variogram fitting is achieved using an automated procedure (Desassis and Renard, 2013).

### 2.1.3 Anamorphosis function fitting

In order to handle the non-Gaussian behavior of the UZD, one possibility is to transform a random function into a Gaussian function using an anamorphosis function fitting such that $\varphi = F^{-1} \circ G$, where $\varphi$ is the anamorphosis function, F the continuous marginal distribution function of unaffected UZD, and G the cumulative density function of the Gaussian score (Chilès and Delfiner 1999). First, the cumulative histogram of the unaffected UZD dataset is established. Therefore, the corresponding Gaussian score is empirically obtained using the frequency inversion of unaffected UZD. The unaffected UZD dataset is transformed into a Gaussian score dataset using an anamorphosis function (Fig1. step 3). This transformation was already used by Flipo et al. (2007) to study aquifer contamination by nitrates.

### 2.1.4 Gibbs sampling - Including soft data

A dry well corresponds to a soft data that can be formulated as constrained by an inequality. One way to deal with these data is to use Gibbs sampling in order to propose a realistic UZD value in accordance with the inequality. The Gibbs sampling method is a way to produce a realization of a Markov random field at a given location (Geman and Geman 1984, Freulon and de Fouquet 1993). This methodology can be directly applied to UZD data (Michalak, 2008) in order to provide a value at each dry well. In this study, Gibbs sampling is applied to the Gaussian score dataset in order to obtain a re-sampled Gaussian score value at each dry well (Fig. 1., step 3). This is made through the distinction between dry well bottom levels (soft data) and UZD measurements (hard data). The UZD measurements are accounted as equality constrained samples and dry well bottom levels are accounted as inequality constrained samples, constituting a lower limit for UZD value, or in other words a minimum value of UZD at the well location. For each dry well a potential value is calculated from successive simulations that reproduce a conditioned value of UZD matching the data distribution and the inequality constraint.

At the end of the Gibbs sampling, the dry well bottom levels are replaced by a probable UZD value at dry well location. This procedure leads to the constitution of a re-sampled Gaussian score dataset.

### 2.1.5 Conditional simulations

The next step is the spatialization of the Gaussian score dataset using geostatistical simulations. The simulation of a random function is the calculation of a possible distribution that matches the variogram and the histogram and that honors the data (Journel, 1986). In this study, the simulation is conditioned by the Gaussian score dataset and is performed on a grid covering the study area using the Turning Bands method (Matheron, 1973). The used variogram model is the same than the one used for Gibbs sampling. Once the simulation is calculated, the resulting Gaussian score map is backtransformed into a UZD map.

One hundred conditional simulations are performed for the calculation of the first guess of the water table map.

### 2.1.6 First guess of the water table distribution

Each Gaussian spatial distribution is backtransformed into a UZD spatial distribution. A preliminary map is obtained averaging the 100 conditional UZD distributions. The first guess map of the water table is obtained deducing this preliminary UZD map from the reference DEM (Fig. 1, step 4).

## 2.2 Water table mapping accounting for uncertain SW-GW connectivity

The second part of the mapping methodology is the final mapping of water table, with the consideration of the SW-GW connection status: the connection status is evaluated for each cell located below the river network using a new disconnection criteria.

### 2.2.1 Defining a disconnection criteria at the reach scale

Stream-aquifer systems fluctuate from a hydraulically connected to a disconnected state due to the development of an unsaturated zone below the stream bed~~associated with a~~. During the switching between connection status, the SW-GW connection status is considered as a transitional state, this condition can occur when the capillary zone intersects the riverbed (Brunner et al., 2009). The disconnected SW-GW condition can occur under different settings such as in case of high hydraulic conductivity contrast between the clogging layer and the aquifer (Brunner et al., 2009; Peterson and Wilson, 1988), the lowering of the water table ~~(Brunner et al. 2009; Peterson and Wilson 1988; Rivière et al. 2014; Wang et al. 2011) caused by natural dry conditions or by permanent groundwater pumping (Dillon and Liggett 1983; Osman and Bruen 2002; Fox and Durnford 2003). The~~ (Dillon and Liggett, 19 or the biological clogging of the riverbed (Newcomer et al., 2016, 2018; Xian et al., 2019). Considering a constant river water level and river width, the disconnection occurs when any further increase of the hydraulic head difference between the water table and the river water level does not affect the infiltration rate from the stream to the underlying aquifer, which remains constant. Wang et al. (2011) and Rivière et al. (2014) proved that the disconnected state is reached when the saturation profile between the riverbed and the water table is stabilized. The saturation profile fills the space between an inverted area below the riverbed and a capillary fringe above the water table (Rivière et al., 2014; Wang et al., 2011). ~~We~~ In the methodology, we assume that the disconnection state is reached when these two capillary fringes are separated without occurrence of a clogging layer. The thickness of these two areas is controlled by the capillary effect which mainly depends on the lithology of both the riverbed and the aquifer. Gillham (1984) proposed values for capillary fringe heights for several lithologies resulting from experimental measurements (Tab. 1). The disconnection criteria is defined as the distance between the riverbed and water table above which the river water and the groundwater are disconnected. It means that for higher distances, a saturation profile develops between the inverted area below the riverbed and the capillary fringe overlying the water table. Accordingly, the disconnection state is identified for a given lithology at each river cell of the estimation grid, when the difference between the first guess water table and the riverbed elevation equals or exceeds an empirical disconnection criteria. The methodology therefore requires either an explicit bathymetric description of the river or an estimation of the riverbed elevation.

**Table 1.** Values for the capillary fringe height, regarding the lithology, after Gillham (1984)

|  | Sand | Silt | Clay |
|---|---|---|---|
| Height of capillary fringe (m) | 0.1 - 1 | 1 - 10 | >10 |

~~Even though knowing the~~ Starting from the knowledge the riverbed lithology, the disconnection criteria can be estimated from Gillham (1984), this ~~value is submitted to a large uncertainty. Indeed,~~ first guess is uncertain given that the distribution of sedimentary heterogeneities into the alluvial plain induces important lithological contrasts (Jordan and Pryor 1992; Flipo et al. 2014) and characterizing such heterogeneities requires important geophysical surveys that are out of reach for the development

of our methodology. At a station, lithology is hence uncertain and a fortiori even more uncertain along a river reach. ~~Another challenge is that~~ However, the disconnection criteria is defined as a threshold difference value between measured UZD and river water level from which the SW-GW connection status ~~varies over time~~switches. In the absence of such criteria in the literature, an optimisation procedure is proposed along ~~a~~ the Seine river network given that piezometers are available in the vicinity of the river and that both in-river water level and water table in the piezometers are recorded synchronously. The optimization

procedure is described into the application section since it is based on the use of temporal data that is not directly required for the mapping methodology.

If the two signals are correlated it indicates that the river and the aquifer are connected. Contrarily, a very low correlation indicates a disconnection. At the reach scale, many piezometers are available. The standardized and normalized hydraulic head and river water level are compared to assess the local connection status of SW-GW. On a scattered plot, the disconnection

appears below a given slope of the regression line.

At a reach scale it is therefore possible to inform the connection status locally (at few stations). Along the river, the distance between the riverbed and the water table is evaluated from the first guess map. The disconnection criteria is evaluated within a range defined by Gillham (1984) as the one that reproduces the most of the locally assessed connection status. In the absence of data, the disconnection criteria defined in our study can be used as a first guess.

**2.2.2  Final step of the mapping methodology**

Disconnected portions of river are deduced from the preliminary water table map with the application of the disconnection criteria. A final dataset of UZD is then created from the UZD first guess at each sampling location, to which connected river sections are added with a nil UZD value. An ordinary kriging is performed with this final UZD dataset (Fig. 1, step 5) for which a variogram model is fitted using the ~~selected UZD data~~ UZD data that is not affected by permanent pumping, as it is described

in section 2.1.2 (Fig. 2., b. and d.). The kriging methodology consists into solving the following two equations system :

$$\begin{cases} Z^* = \sum_\alpha \lambda_\alpha Z_\alpha + m\lambda_m \\ VarR = C_{00} - \sum_\alpha \lambda_\alpha C_{\alpha 0} \end{cases}$$

Where $Z^*$ is the estimation, $\alpha$ is the observation point, $\lambda_\alpha$ and $\lambda_m$ are the weights for observation point and mean value, R is the residual value (i.e. absolute error), $C_{00}$ and $C_{0\alpha}$ are the covariance function values for the origin and the $\alpha$ point. Therefore, the value of R is not determined solving this system, only the variance of R is calculated.

The final water table map is obtained using the reference DEM, from which the UZD kriged map is deduced.

## 3 Results – Water table mapping of Paris urban area

The methodology is demonstrated on the Paris urban area. This urban area covers 900 km$^2$ and includes Paris city and its closest peripheral suburbs.

### 3.1 Paris urban area

The Seine and Marne rivers constitute a meandering fluvial system flowing from the South-East to the North-West (Fig. 3). The confluence between the Marne and the Seine River is located in the South-East of the studied area.

The alluvial plain of the Seine and Marne rivers is overlying incised valleys of Eocene to Oligocene sedimentary series, exposing late Lutetian limestones in the south of the study area and early Bartonian limestones in the north (Fig. 3). Alluvial sediments constitute the alluvial aquifer and substratum of the Seine and Marne rivers. Lutetian and Bartonian limestones are underlying aquifers which are separated by thin heterogeneous and discontinuous formations with low hydraulic conductivity. The high proportion of soil sealed areas and the proliferation of pumping wells due to the urbanization, reduces the infiltration of rainfall, making the ~~anthropic~~ anthropogenic pressure the main controlling factor of water table and SW-GW connection status.

Water table have been monitored by water managers since the 1970's in central Paris area and since the 2000's in suburb areas. Water managers noticed that water table of alluvial aquifer in the central area is usually stable at very low levels such that drying out of superficial aquifers may occur. The water table of peripheral areas remains unaffected by such water table drawdown.

Regardless of the groundwater context, the Seine river is fully embanked, and the river bottom is periodically dredged for navigation purposes along the Paris city crossing. Given those anthropogenic forcings, water managers suspect that the Seine river may be disconnected from its underlying alluvial aquifer in some parts of Paris central area. The ~~study~~ interpretations of the join response of alluvial aquifer to hydrological events during the 1990-2018 period ~~is~~ described further are supported by monthly records for UZD at monitored piezometers located nearby the Seine river. Seine river water levels vary under two different hydrological regimes (Fig. 4.a.): one nominal hydrological regime, corresponding to the low flow periods during which the river flow and water level are artificially regulated for navigation and water management purposes, and one flood

**Table 2.** Overview of the used samples for the application of water table mapping

|  | Total number of wells | Number of wells affected by pumping wells | number of dry wells |
|---|---|---|---|
| LWC | 314 | 25 | 47 |
| HWC | 202 | 33 | 11 |

regime during which the river water level reaches the flood peak (eventually causing flood damages). The UZD can vary in relation with the Seine river water level or not. In the case of no variation of UZD, it is assumed that UZD is regulated by the GW pumpings, inducing a disconnection between water table and river water level.

### 3.2 UZD datasets: Low and High flow campaigns

This study is based on two UZD snapshot campaigns (Tab. 2) involving measurements in piezometers that are not periodically monitored (Fig. 3): the low water campaign (LWC), that gathered 314 measurements during the low flow period of October 2015, and the high water campaign (HWC), gathering 202 measurements during the June 2016 flood event. Both campaigns last about a week. HWC took place a few days after the flood peak (1750 $m^3.s^{-1}$) was reached at the Parisian Austerlitz gauging station. The two datasets include around 22 % of samples affected by anthropic pressure (pumping wells and underground

structure) (Tab. 2). Most of these samples are located in the Paris central area where the water table is affected by permanent pumping.

### 3.3 Reference DEM for each campaign

The used DEM is the IGN scan 25 (IGN, 2015). As previously mentioned, the DEM is first merged with hydrological data specific for each campaign. Then it is smoothed using a 325 m research radius for moving average filtering.

River water levels are deduced from the recorded data of six discharge gauging stations. The distribution of river water levels is interpolated using a constant gradient between each gauging station. During low flow period, the average gradient value is 0.01‰. The Seine River discharge is regulated through a series of locks and dams for navigation purposes. At each lock station, water levels are maintained at a given elevation below a threshold water flow of 600 $m^3.s^{-1}$ at the Paris Austerlitz station. When a flood occurs as in June 2016, the lock stations are opened, and the water surface returns to its natural 0.2 ‰

gradient. During the LWC, the Seine river discharge was 160 $m^3.s^{-1}$, so that all locks were up, while they were ~~down~~ open during the HWC, when the average discharge still reached 1000 $m^3.s^{-1}$ a week after the flood peak.

### 3.4 Variograms

The experimental variograms, and the associated fitted variogram models are depicted (Fig. 2). For both datasets, the shape of the variograms is similar with a sharp increase of the semi-variance nearby the origin, followed by a smooth evolution until it

reaches the sill value. The range is higher for the LWC than for the HWC. The range of the Gaussian score is 12 km for LWC and 5 km for HWC (Fig. 2.a. & b.). The range of the raw data is 2 km for HWC while it is 6 km for the LWC. It can be noted that the variographic models for unaffected UZD data differ between LWC and HWC datasets in terms of sill value (Fig. 2.a. & b.). The sill value for the variogram model of the unaffected UZD HWC dataset is 8 $m^2$ while it is 5 $m^2$ for the unaffected

UZD LWC dataset. This can be due to either the lower number of samples collected during the HWC, or to variations in the inner structure of the flow propagation process (Chen et al., 2018; Samine Montazem et al., 2019). For both campaigns (LWC and HWC), the variogram models of Gaussian scores have a range larger than the one of unaffected UZD datasets. This is due to the increase of the spatial correlation of the variable once the unaffected UZD data is transformed into Gaussian data.

## 3.5    Assessing the disconnection criteria

The Gaussian simulations are run on a 25 m x 25 m grid that matches the DEM resolution. The average of the hundred UZD values is subtracted to the smoothed DEM that includes river water levels evaluated for each hydrological context (LWC and HWC). The streambed of the Seine river consists of mixed fine sand, and silt. The a priory capillary fringe height is comprised within 0.1 m and 1.0 m (Tab. 1). Therefore, the a priory value for the disconnection criteria is comprised between 0.2 m and 2 m. The available observed data is composed of monthly measurements of UZD among 26 piezometers during the 1990-2018

period. These piezometers are distributed along 18 cross-sections of the Seine river. For each piezometer, standardized UZD and river water level values are calculated. As described in section 2.2.1, the SW-GW connection status can be deduced from the relation between UZD and river water level. Two ~~class~~ classes of piezometers are identified given the linear regression between standardized UZD and standardized river water level: disconnected piezometers and connected piezometers. Please note that during disconnection, the flow rate is still related to the hydraulic head difference. The transition cases are therefore

included in the connected piezometer group. In case of a significant slope of the regression line (>0.57), the piezometer is considered connected. It is disconnected otherwise. ~~The piezometers are classified depending on the linear regression results.~~ ~~The discriminated slope value between the connected and disconnected piezometers is 0.57.~~ 15 piezometers are considered as connected and 11 piezometers are considered as disconnected. Therefore, 9 cross-sections along the Seine river are connected and 9 sections are disconnected (Fig. 5.a.).

As an example, two contrasted situations among the 26 piezometers are displayed (Fig. 4.a.). The UZD measured in the blue piezometer ~~evolves at the same time as~~ is linearly related to the river water level, while the UZD measured in the red piezometer remains roughly constant. There is a linear regression between UZD measured in the blue piezometer (Fig. 4.b.) which confirms that the blue piezometer is connected. In the case of disconnected piezometers, a constant UZD value is measured for most samples. It indicates that UZD is regulated artificially (Fig. 4.a.).

## 3.6    Sensitivity analysis of the disconnection criteria

The distribution of SW-GW connection status is constrained by the disconnection criteria. To estimate this criteria, a sensitivity analysis is achieved. The tested values range from 3 m to 0.5 m with a 0.1 m step. This analysis shows that it is not possible to validate the connection status for all cross-sections. Therefore, we compare the ~~number~~ relative numbers of

matched connected cross-sections and disconnected cross-sections. When the ~~number is~~ relative numbers are equal, the optimal value is reached, ~~minimizing the overestimation of disconnection and connection.~~ maximizing the total number of sections for which the connectivity status is correctly predicted. Different methods to evaluate the best criteria can be used (e.g. only valid disconnected cross-sections or valid connected cross sections). In this study the maximization of relative number of connected and disconnected sections is used in order to obtain an average value of the disconnection criteria, that does not favor either disconnection or connection. A way to obtain a better validation of cross-section would be to spatialize the disconnection criteria. However, the spatialization of the disconnection criteria must be supported by geological arguments. This optimal value is 0.75 m (Fig. 5.b.). This value is used to obtain the final water table maps. The value of the disconnection criteria impacts the length of disconnected reaches. When the value for disconnection criteria is overestimated, the length of disconnected reach is underestimated. Contrarily, when the value is underestimated, the length of disconnected reaches is overestimated. In the application presented here for LWC, the length of disconnected reach for a 3m disconnection criteria value is 150 m in the central area, while it reaches a 6 km length when the disconnection occurs for a 0.25m disconnection criteria. When the optimal value of 0.75 m is applied for disconnection criteria, the length of disconnect reach is 5 km.

Further investigation could be carried out to evaluate the reliability of the estimated disconnection criteria, comparing it with the application of other methodologies such as it is described in Lamontagne et al. (2014). Thought this would allow for the determination of SW-GW flowrate and hydrogeological dynamics, it cannot be applied into our case study context given that there is no data about the riverbed hydraulic conductivity. Such development would constitute a supplementary step after the water table mapping toward the description of the hydrological functioning of the study area.

## 3.7 Final mapping integrating SW-GW connectivity

The most important GW hydraulic gradient (1 %) are located close to the Seine and Marne rivers and the areas with an important topographic gradient (Fig. 6). The lowest values of hydraulic gradient are comprised between 0.1 ‰ and 1 ‰ with an average 0.6 ‰ value in rather flat alluvial plains in the north area and the south-east area. The global flow pattern is therefore driven by SW-GW connection status and topography, at the exception of the central area where permanent pumping generates significant water drawdown and a subsequent SW-GW disconnection. In this area, the difference between riverbed elevation and estimated water table is 4 m. The implementation of disconnected reach during final mapping is a key element to reflect the specificity of urban groundwater such as water drawdown caused by pumping wells. The mapped water table nearby disconnected reach is only affected by the observed depletion of water table in wells and dry wells. All disconnected sections are located in the central area during both campaigns. The rise in river water levels during HWC modifies the water table map significantly, especially in the vicinity of the river.

The SW-GW relation type for connected sections (gaining, loosing and asymmetrical) is established regarding the head difference between the river water level and water table at a 50 m distance (two pixels of the map) from the river center-line. In cases where the river water level is higher that water table for both river banks, the river is loosing water toward the aquifer. In the opposite case, the river is gaining groundwater. The sections where the river is gaining on one bank and loosing on the other were also identified.

**Table 3.** Length of disconnected sections, connected gaining sections and connected loosing sections and head difference between water table and riverbed calculated from preliminary mapping, for LWC and HWC

|  | LWC | HWC |
|---|---|---|
| Disconnected length (km) | 4.9 | 0.8 |
| Loosing sections length (km) | 30.3 | 20.0 |
| Both gaining and loosing (km) | 38.3 | 39.3 |
| Gaining sections length (km) | 11.5.4 | 25.0 |

The main effect of ~~such hydrological events though the increase of the hydraulic head is to favor river infiltration towards~~ flood events is to increase the river water level. In connected sections, the increase in hydraulic gradient between river and water table favors the river infiltration toward the aquifer. Comparing Fig. 6.a with Fig. 6.b, it appears that this infiltration causes the switching from loosing to gaining SW-GW relation type during flood events. This is supported by the reduction of the total length of loosing sections during LWC (Tab.3). The hydrogeologic flow associated with the increase in infiltration induces the reconnection process propagating from the loosing sections toward the disconnected sections.

As a consequence, almost the whole river network is reconnected to the GW, leading to a rise of the mapped water table. As pumping is increased during a flood to avoid damages against the buildings and underground infrastructures, a small portion of the Seine River remains disconnected in central Paris (0.75 km, see Fig. 6.b.)

## 4  Conclusions

This study demonstrates an application for an innovative and generic mapping methodology of the water table in an urbanized alluvial environment. Besides accounting for information brought by the knowledge of dry well locations and depth, the methodology introduces a SW-GW disconnection criteria for the first time in water table mapping.

The methodology is demonstrated for the case of the Paris urban area, for which it confirms GW managers suspicion for a disconnection between SW and GW Downtown Paris. Indeed, the water table appears to be locally depleted causing SW-GW disconnection with the alluvial aquifer. Water table maps lead to the identification of spatialized SW-GW disconnected portions in the central area of the city. In the case of connected SW-GW, an important hydraulic gradient is observed in the vicinity of the river. In the case of a disconnected state, the water table remains unaffected by the hydrographic network and follows the natural slope of the DEM. Such methodology offers the opportunity of an automated water table mapping connected with GW monitoring network in urbanized areas exposed to flood risk.

*Acknowledgements.* This study has been carried ~~out thanks to~~ with the support of the Programme d'~~actions de prévention des inondations~~ Actions de Prévention des Inondations de la Seine et de la Marne ~~Franciliennes~~ franciliennes (PAPI SMF) steered by the Etablissement

public territorial de bassin Public Territorial de Bassin Seine Grands Lacs (EPTB SGL, the Seine watershed basin institution for river water flow regulation). The used synchronous datasets was were produced and kindly provided by the Inspection générale des carrières Générale des Carrières (IGC, Mairie de Paris) and the other stakeholders involved in that program (Société du Grand Paris, Conseil départementaux de Seine Saint-Denis et du Val de Marne, RATP, CEREMA). We want to address special thanks to A.M. Prunier-Leparmentier and S. Ventura-Mostacchi (IGC, Mairie de Paris) for their constant commitment in the management of and advices on water table monitoring in Paris city. This monitoring is a key point for the analysis of piezometric head time-series. Their technical expertise about Paris city groundwater was also a major contribution to this publication. This work is also a contribution to the PIREN Seine research program on the Seine basin, part of the Long Term Socio Ecological Research (LTSER), french infrastructure Zone Atelier Seine. Finally, we kindly thank Graham E Fogg for supervising the review process, as well as the two anonymous reviewers for their accurate comments, which helped improve the paper significantly.

**Figures**

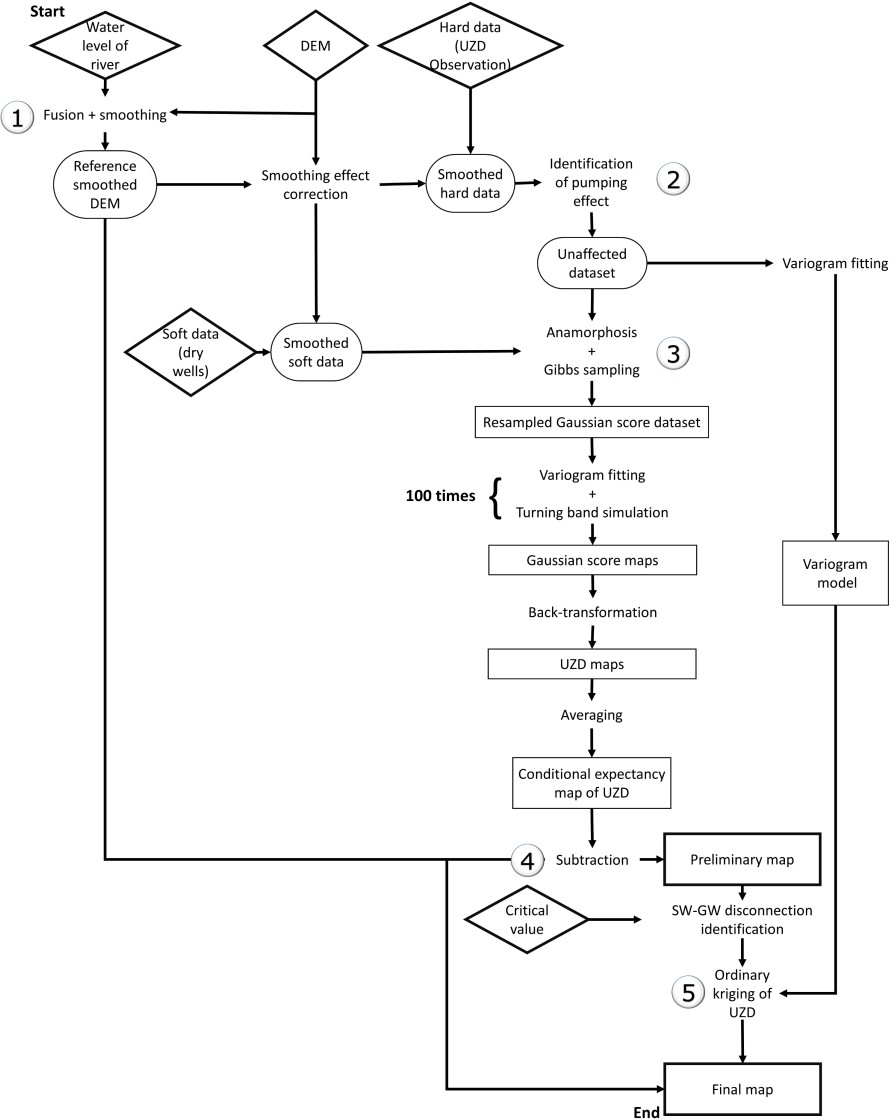

**Figure 1.** Flow chart for the mapping of water table. Steps 1 to 4 for first guess map. Step 5 for the final map. UZD: unsaturated zone depth. Diamonds display raw data, ellipses display input data after pre-processing and squares display intermediary products.

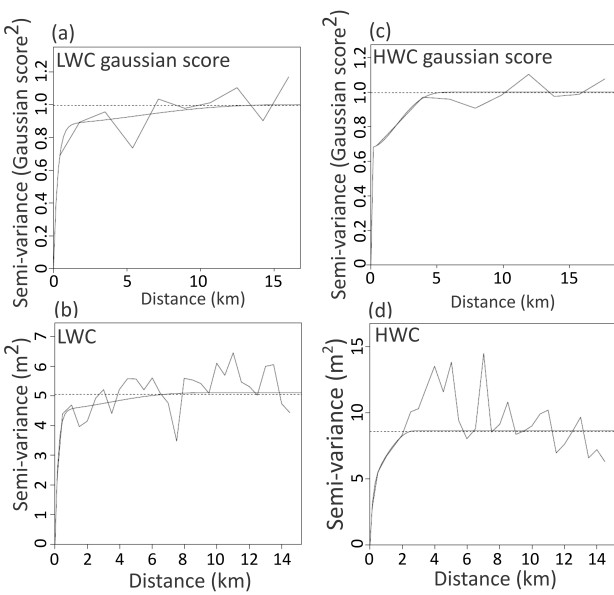

**Figure 2.** Experimental variogram and fitted variogram model of unaffected UZD data and Gaussian score, for LWC dataset in the left column (a., b.) and HWC dataset in the right column (c., d.)

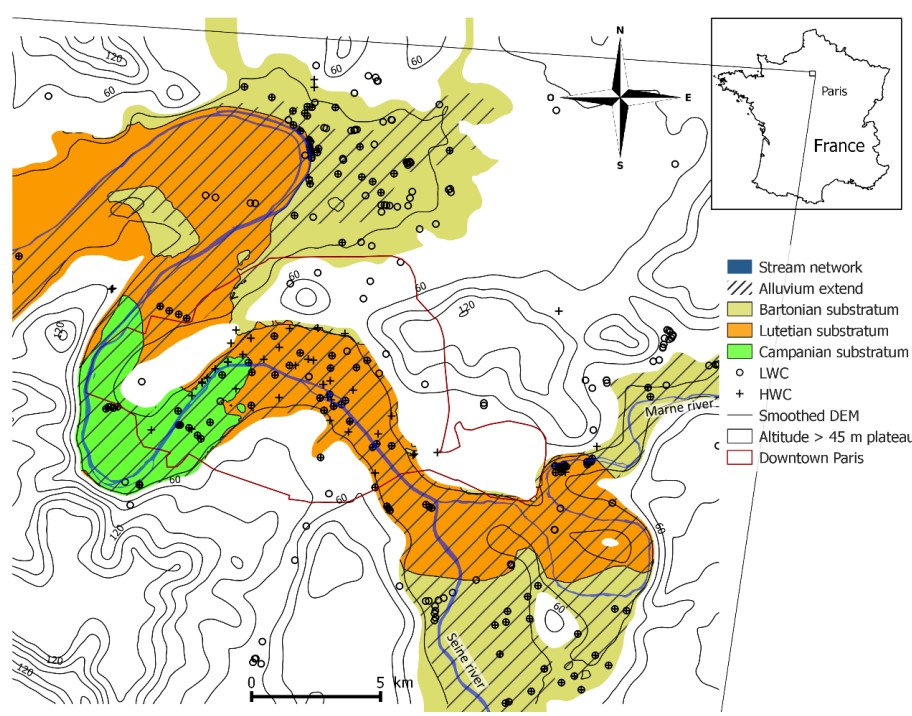

**Figure 3.** Alluvial plain and regional substratum of the Paris urban area. piezometers location for the two campaigns: low water campaign (LWC) and high water campaign (HWC).

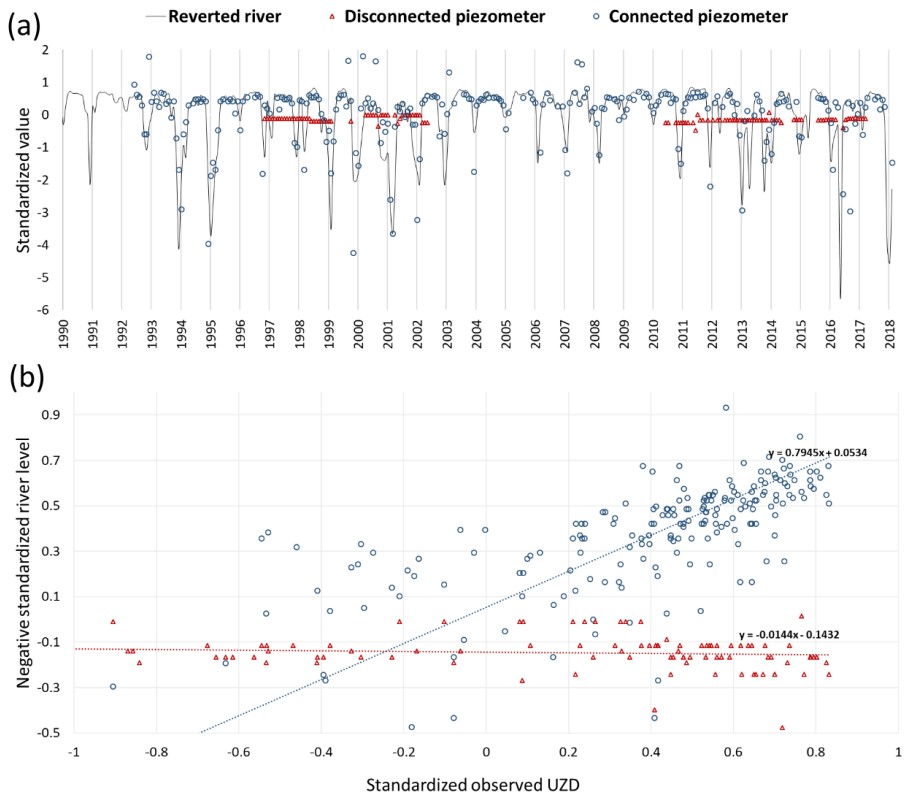

**Figure 4.** Temporal analysis charts for SW-GW connection status determination: (a) recorded time-series for river, disconnected piezometer and connected piezometer; (b) relationship between standardized river level and UZD for disconnected piezometer and connected piezometer.

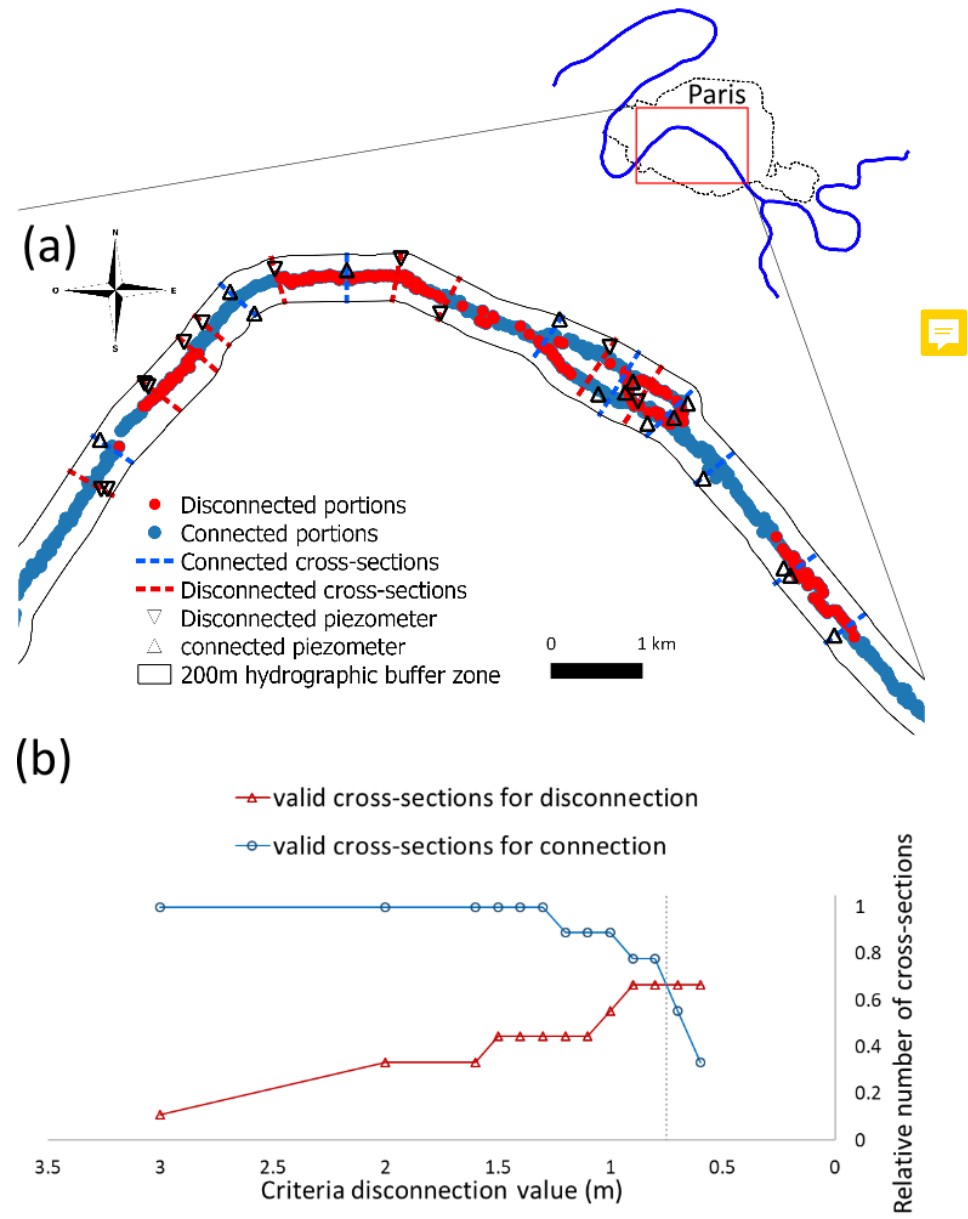

**Figure 5.** Graphical representation for disconnection criteria adjustment: (a) map of observed SW-GW status related to estimated SW-GW connection status using the optimal 0.75 m value for disconnection criteria; (b) Relative number of valid SW-GW connection status out of 9 disconnected cross-sections and 9 connected cross-sections.

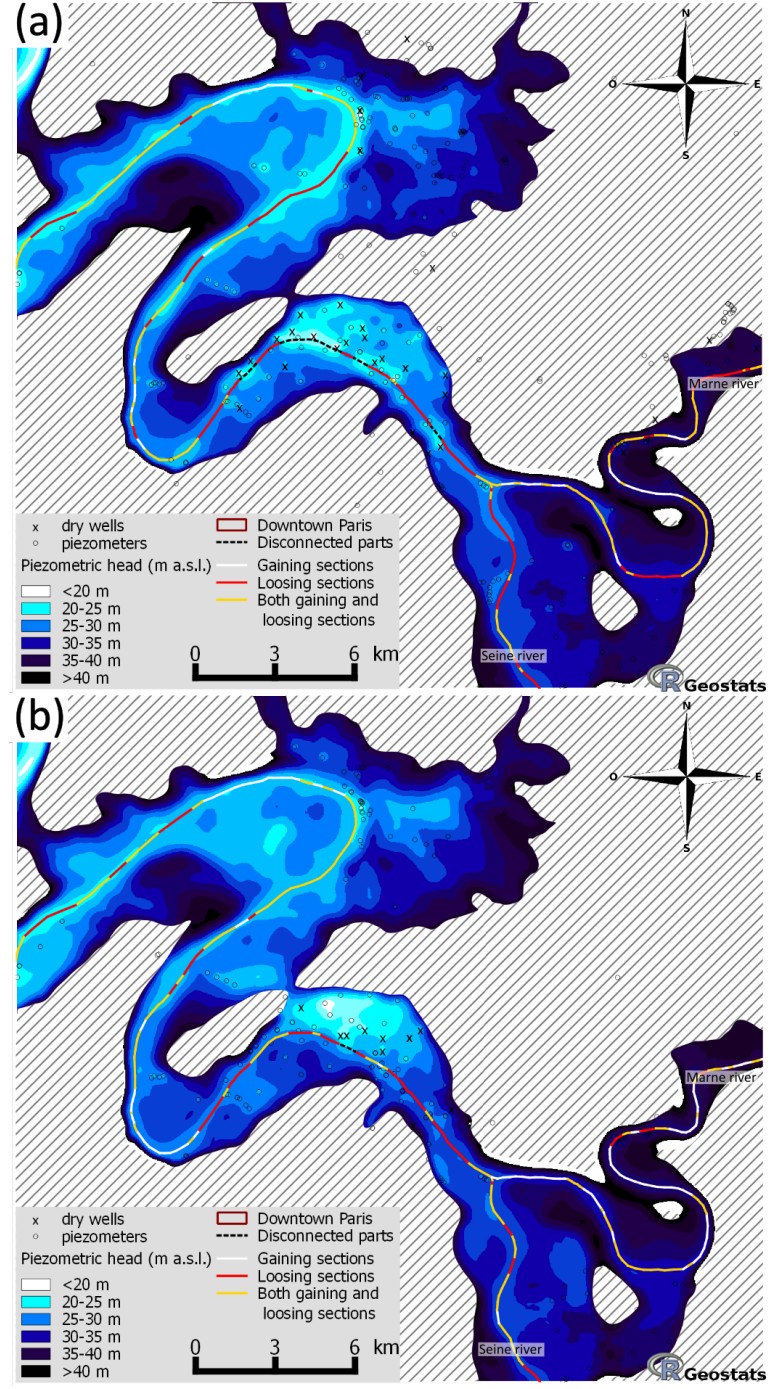

**Figure 6.** Final water table maps obtained using ordinary kriging of UZD deduced from the reference DEM for a) LWC and b) HWC. The disconnected reaches of the river network are indicated in red for a disconnection criteria of 0.75 m

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
