# Peer review of "Technical note: Water table mapping accounting for river-aquifer connectivity and human pressure"

_Hydrology and Earth System Sciences, 2019_

## Referee Comment (RC1) · Anonymous Referee #1 · 3 Jun 2019

General comments ——————

This paper presents a rather sophisticated interpolation methodology for water table mapping. The novelty lies in accounting for a priori-unknown groundwater-surface water connectivity and dry well information. I agree that these aspects can be important for water table mapping. The study seems thorough and will certainly be of interest to the readership of HESS.

On the downside, I find that the paper lacks clarity at numerous occasions (see my detailed comments below). It also appears that questions (i) and (iv) stated at the end of the introduction are left without an answer. These questions call for a comparison

of different methods (or different levels of refinement), but no such comparison is presented. I think it would be very informative to do so; indeed, without such a comparison, the benefits of the various refinements are not obvious.

I think that this work can make a great contribution to the literature after these comments are addressed in the context of a moderate revision.

Detailed comments ——————————

P1 L14: Suggest adding reference to a classic publication (e.g. Winter et al. [1998]) to support the first sentence.

P1 L14-15: This sentence is confusing. In this case, the water table is not below the riverbed: it is at the river water level.

P1 L19-20: Suggest also mentioning topography as a controlling factor and adding reference to Bresciani et al. [2016].

P2 L3: What do you mean by "usual estimators"?

P2 L.-16: Also note that Bresciani et al. [2018] obtained good results with the diffusion kernel interpolation method.

P2 L23: Shouldn't it be simply "large uncertainty in the estimation" instead of "large standard deviations of the estimation errors" (this would be the error of the error...)? Same comment on L27.

P3 L5: What "drawback" are you referring to? Obviously, the water table is always largely controlled by recharge. I do not see what point is being made here.

P3 L12: What do you mean by "define the SW-GW connection status"? Maybe you rather mean "determine the SW-GW connection status"?

P3 L20: What does "the dataset analysis" refer to?

Figure 1: Arrows are missing. This makes the figure difficult to understand.

Figure 1: What is "Water profile of river"? Do you mean river water level?

P4 L1: On the previous page you refer to Gaussian statistics, and here non-Gaussian. This is confusing.

P4 L2: Reference needed. Also note that this is not rigorously true. Namely, Desbarats et al. [2002] suggested that this assumption is not appropriate at the scale of a single catchment.

P4 L5-7: Please explain the reason for smoothing the DEM, and clarify the sentence "the search radius is relevant with the average width value of the stream network".

P4 L8-13: Unclear.

P4 L20-21: The value of 10 m seems totally arbitrary. How did you come up with this value? How does the choice of this value impact the results? Furthermore, I did not understand what you did with this first category of data.

P4 L23-24: Should I understand that you refer to the Gaussian score as a variable? This is unclear.

P6 L22: And riverbed thickness?

P6 L23-24: "is submitted to": I guess you mean "is subject to".

P6 L26: "At a station": I guess you mean "At point scale".

P6: L28: What does "such criteria" refer to? Same for "optimization procedure".

P6 L23 – P7 L6: I am getting a bit lost here: which method did you use in the end? I think I ultimately understood, but the organization of ideas could be improved.

P7 L11: What are the "selected UZD data"?

P7 L11: Fig. 5 is referred to before figures 2-4; please correct this.

Fi.g2: Indicate river names on the map.

P8 L7-8: "The study of...": What study are you referring to?

P9 L 6: Not sure about the meaning of "up" and "down": this is confusing. Suggest using "open" and "closed" instead.

P9 L29: Why 0.57? This seems totally arbitrary.

P9 L30-31: Repetition of previous sentence.

P10 L10-11: I think it would make more sense to maximize the total number of sections for which the connectivity status is correctly predicted.

Figure 4b: Out of how many cross sections in total?

Figure 4: (b) is not announced in the caption.

P10 L28-29: "river infiltration towards the aquifer": This does not seem to be supported by the results, which suggest that groundwater flows from the aquifer towards the river (Figure 6b).

References ————

Bresciani, E., P. Goderniaux, and O. Batelaan (2016), Hydrogeological controls of water table-land surface interactions, Geophysical Research Letters, 43, 9653-9661.

Bresciani, E., R. H. Cranswick, E. W. Banks, J. Batlle-Aguilar, P. G. Cook, and O. Batelaan (2018), Using hydraulic head, chloride and electrical conductivity data to distinguish between mountain-front and mountain-block recharge to basin aquifers, Hydrol. Earth Syst. Sci., 22(2), 1629-1648.

Desbarats, A. J., C. E. Logan, M. J. Hinton, and D. R. Sharpe (2002), On the kriging of water table elevations using collateral information from a digital elevation model, Journal of Hydrology, 255(1-4), 25-38.

Winter, T. C., J. W. Harvey, O. L. Franke, and W. M. Alley (1998), Ground water and surface water - A single resource, U.S. Geological Survey Circular No. 1139, U.S.

Geological Survey, Denver, Colorado.

---

## Referee Comment (RC2) · Anonymous Referee #2 · 24 Jun 2019

This manuscript evaluates the connection status of an urban river through the statistical generation of a water table map. Determining groundwater - surface water interactions at the larger scale for rivers is a current research topic and thus the manuscript is well suited for the journal. In general, I have found the paper interesting and potential quite useful with some adjustments. However, I cannot comment on the methodology used for generating the water table map because I have no expertise in the methods used. The manuscript will require some English editing but this is a minor concern.

My main comment is that the authors could have also used simpler alternative approaches to evaluate connectivity to compare with their methodology. Likewise, some

of the hydrogeological assumptions used to derive the map are probably incorrect. I will explain these concerns below.

1) Connectivity state. The authors are probably aware but have failed to mention that there are three hypothesised connectivity states for a river - alluvial aquifer system: Connected (gaining or losing), disconnected, and transitional. The transitional state was not described here and needs to be mentioned in the context of the system (transitional conditions can occur when the capillary zone intersects the riverbed. The concept of the riverbed clogging layer and the necessity of its presence to generate disconnected conditions under most field conditions in a temperate climate needs to be described.

2) Hydrogeology in an urbanised area: I was uncomfortable with some of the assumptions used in the generation of the water table map, in particular that there is no recharge in urban areas due to the presence of 'impervious' structures. This is not consistent with findings elsewhere, which have shown that 'impervious' structures like road networks are never completely so. Moreover, most large old cities have large old and leaky water distribution and sewerage networks. Whether this is the case for the study area I do not know but evidence must be provided to satisfy that no recharge occurs there. In addition and of consideration for the generation of the water table map, sewerage networks can also act as drains and thus impart an upper limit for the position of the water table.

3) Hydrogeological interpretation: It is probably necessary to compare the assessment made using a water table map with a more hydrogeologically-based interpretation. For example, Lamontagne et al. (Hydrological Processes 28: 1561-1572) provide a methodology where assessments of connectivity for rivers can be made using pairs of surface and groundwater level measurements for different hydrogeological properties of riverbeds and aquifers. Even though the authors may not have all the information required (e.g. riverbed hydraulic conductivity) the Lamontagne et al. approach can still be used in a sensitivity analysis context. Indeed, a sensitivity analysis should also be

applied to the water table approach as well if possible - what is the potential error in the map?

The empirical approach used by the authors to evaluate connectivity (Fig. 3) could be flawed based on hydrogeological principles. The lack of response of a water table to variations in surface water level could be due to a number of factors other than a river being disconnected. Indeed, even when disconnected, the specific recharge rate and flux below a river will increase at higher water levels (due to a greater head and wetted area), thus a response of the water table is still possible. On the other hand, a river could be connected but the alluvial aquifer have a very low transmissivity, which could result in a subdued variation in the water table especially if bores are at some distance from the river or the variation in river stage only for a short period of time. At a very low transmissivity, the response will only be notable when very close to the river. For a given change in river level, there will be a certain distance where the water table response can be practically measured considering other sources of variation.

Minor comments:

P2, Line 1:'Embankments' should probably be 'levees' if they raise the water level at which water would spill into a floodplain. P4. Use of unsaturated zone thickness: I did not understand why this was better than using the water table elevation. It would also be preferable to use the term 'vadose zone' instead of 'unsaturated zone' considering capillary effects are of interest. P.20. Basis for the 10 m threshold. Some additional justification is required for using this threshold to identify disconnected conditions. Drawdown cones can be much deeper, especially when transmissivity in the aquifer is low. P.6, line 11. This is only correct if the river level is constant. If the river level increases, the specific recharge rate and overall recharge flux below the river will increase because of a higher hydraulic head and a larger wetted area (i.e. wider river). Figures: Could in general all be improved - very faint lines and symbols in particular. Figure 6a was interesting and should be complemented with similar 2D cross-sections when feasible to get an idea of the shape of the water table near the river.

---

## Author Comment (AC1) · 23 Jul 2019

We thank the reviewers for their valuable review of our manuscript and for their constructive comments, which substantially helped improving the quality of the paper. Please find hereafter a point-by-point rebuttal with a new version of the manuscript (supplementary pdf file) where the correction has been made. Changes to the text in the manuscript are highlighted.

"It also appears that questions (i) and (iv) stated at the end of the introduction are left without an answer. These questions call for a comparison of different methods (or different levels of refinement), but no such comparison is presented. I think it would

be very informative to do so; indeed, without such a comparison, the benefits of the various refinements are not obvious."

This is a relevant point raised by the referee N°1. We propose to change the first addressed question of the introduction to "(i) what are the appropriate steps for water table mapping of superficial aquifer?"

We think the comment pertaining to the fourth question meet up with the second referee's comment which is about the interpretations of the produced map in terms of SW-GW flow rate and groundwater flow dynamics. These aspects are not developed in this study. We believe that such a development should constitute further works of the application of this study that we want to keep focus on the mapping issue.

List of changes:

P1 L14: Suggest adding reference to a classic publication (e.g. Winter et al. [1998]) to support the first sentence. => P1-L15: adding winter et al. 1998 reference

P1 L14-15: This sentence is confusing. In this case, the water table is not below the riverbed: it is at the river water level. => P1-L15: indeed there is no need to mention the riverbed, removing the confusing "below the riverbed"

P1 L19-20: Suggest also mentioning topography as a controlling factor and adding reference to Bresciani et al. [2016]. => P1-L20: adding Bresciani et al. 2016 reference

P2 L3: What do you mean by "usual estimators"? => P2-L4: change "usual estimators" to "linear estimators"

P2 L.-14: Also note that Bresciani et al. [2018] obtained good results with the diffusion kernel interpolation method. => P2-L15: adding Bresciani et al. 2018 reference

P3 L5: What "drawback" are you referring to? Obviously, the water table is always largely controlled by recharge. I do not see what point is being made here. => We thank you to point out this mistake which is a remaining artifact resulting from former

version of the study. We propose to remove the two sentences since the methodology does not require to identify or quantify recharge.

P4 L2: Reference needed. Also note that this is not rigorously true. Namely, Desbarats et al. [2002] suggested that this assumption is not appropriate at the scale of a single catchment. => P2-L26/28: adding one sentence about the topography - water table correlation and corresponding Haitjema and Mitchell-Bruker 2005 reference

P3 L12: What do you mean by "define the SW-GW connection status"? Maybe you rather mean "determine the SW-GW connection status"? => P3-L11: change "define" to "determine"

P3 L20: What does "the dataset analysis" refer to? Figure 1: Arrows are missing. This makes the figure difficult to understand. Figure 1: What is "Water profile of river"? Do you mean river water level? => P3-L19/20header: changed "Firstly, the dataset analysis is achieved in order to constitute the raw dataset" to "Firstly, the raw dataset is composed of each measured unsaturated zone depth for the corresponding measurement campaigns" => Arrows were added to Fig.1 between each item.

P4 L5-7: Please explain the reason for smoothing the DEM, and clarify the sentence "the search radius is relevant with the average width value of the stream network". => P4-L7/13: adding explanations for the DEM smoothing and the smoothing effect correction: " The smoothing of the DEM is required to avoid the occurrence of high frequency topography signals that would not be relevant with the water table signal. The search radius is in agreement with the average width value of the stream network in order to ensure that the river water level is kept after smoothing. The difference between rough DEM and smoothed DEM may be important in locations where the topographic slope is the most important. These locations include crucial areas nearby the riverbanks. Therefore, this difference is calculated at each sampling points. Due to the use of UZD, this generates a biased estimation of water table at these locations, given that this difference is not yet accounted for into the UZD measured value."

[Figure]

P4 L5-7: Please explain the reason for smoothing the DEM, and clarify the sentence "the search radius is relevant with the average width value of the stream network". => P4-L23-29: adding explanation for the data selection step: "The second category is composed by the other samples. Information about the locations of pumping wells is required to identify these samples. The observed minimal UZD of depreciated areas can be use as a threshold value to differentiate affected points from non-affected points. In this study, the samples with UZD value greater than 10 m are grouped in this category. Note that this value may vary according to the case study. This differentiation is required to elaborate a geostatistical tool (i.e. variogram model) that only depends of natural variability. Therefore, all the variographic studies are performed on this second category called unaffected UZD dataset."

P4 L8-13: Unclear. => P4-L31/33: adding explanation for better comprehension: "The Gaussian score variable used for Gibbs sampling-conditionnal simulation steps is described in next subsections. UZD is the variable ultimately used for ordinary kriging."

P2 L23: Shouldn't it be simply "large uncertainty in the estimation" instead of "large standard deviations of the estimation errors" (this would be the error of the error)? Same comment on L27. => P7-L21/P8-L3: adding ordinary kriging equation system and description. The strictly speaking outcome of kriging are the standard deviations of the estimation errors.

P6 L22: And riverbed thickness? => We assume that there is no vertical stratification between riverbed and aquifer considering that the riverbed is periodically dredged

P6 L23-24: "is submitted to": I guess you mean "is subject to". => changed at P6 L31-32

P6 L26: "At a station": I guess you mean "At point scale". => Point scale refers to a very localized spot, such as a hyporheic zone scale. Here we consider multiple information coupling river hydraulic head and a nearby piezometer. We prefer to call it a station than a point.

**HESSD**

P6: L28: What does "such criteria" refer to? Same for "optimization procedure". => P7-L3/4: Adding criteria description before mentioning it => P7-L7/8: Adding reference to optimization procedure description to the corresponding further section

P6 L23 – P7 L6: I am getting a bit lost here: which method did you use in the end? I think I ultimately understood, but the organization of ideas could be improved. => P6-L31/P7-L4: Improving text readability with previous modifications

P7 L11: What are the "selected UZD data"? => P7-L19-20: Adding a better description of the used dataset, referring to previous section

P7 L11: Fig. 5 is referred to before figures 2-4; please correct this. Fi.g2: Indicate river names on the map. => Figures: moving Fig.5 to Fig.2

P8 L7-8: "The study of...": What study are you referring to? => P8-L22-23: changed word "study of" into "interpretations about . . . described further"

P9 L 6: Not sure about the meaning of "up" and "down": this is confusing. Suggest using "open" and "closed" instead. => P9-L17: changed word "down" into "open"

P9 L29: Why 0.57? This seems totally arbitrary. => The 0.57 value results from a classification of the relation between water table and river water level timeseries among two classes of piezometers of which two representative piezometers are showed into Fig.4.

P9 L30-31: Repetition of previous sentence. => Repetition removed.

P10 L10-11: I think it would make more sense to maximize the total number of sections for which the connectivity status is correctly predicted. => P10-L29: This is what has be done. Thank you to point out the need for clarification.P10/L29-31 has been reworked.

Figure 4b: Out of how many cross sections in total?

Figure 4: (b) is not announced in the caption. => Fig.5: (b) description added

[Figure]

Figures: Fig.6: adding river names, adding new color for connected river and points to avoid confusion between water table color bar and other features of the map, and improve contrast.

Please also note the supplement to this comment:
https://www.hydrol-earth-syst-sci-discuss.net/hess-2019-101/hess-2019-101-AC1-supplement.pdf

[Figure]

**Supplement:**

[revised manuscript text omitted]

---

## Author Comment (AC2) · 23 Jul 2019

We thank the reviewers for their valuable review of our manuscript and for their constructive comments, which substantially helped improving the quality of the paper. Please find hereafter a point-by-point rebuttal with a new version of the manuscript where the correction has been made. Changes to the text in the manuscript are highlighted.

"My main comment is that the authors could have also used simpler alternative approaches to evaluate connectivity to compare with their methodology. "

[Figure]

=> We agree with the main comment pointing that the determination of SW-GW connection status could be improved using the methodologies mentioned further in the comment, especially to estimate the dynamics and the flow rate between stream and aquifer. However, our paper focuses on the water table mapping, only, which is a fundamental information for hydrogeologists, who derive from it, under given assumptions, such as permeability values, dynamics of the system and flow rate estimates. We choose to consider only hydraulic head and differential hydraulic head in order to set the boundary conditions of the water table mapping. Flow rate estimates between stream and aquifer will be part of a study describing the hydrological functioning of the Paris city area.

"1) Connectivity state. The authors are probably aware but have failed to mention that there are three hypothesised connectivity states for a river - alluvial aquifer system: Connected (gaining or losing), disconnected, and transitional. The transitional state was not described here and needs to be mentioned in the context of the system (transitional conditions can occur when the capillary zone intersects the riverbed. The concept of the riverbed clogging layer and the necessity of its presence to generate disconnected conditions under most field conditions in a temperate climate needs to be described.

=> It is true that transitional state and clogging layer need to be described in our paper. We added some modifications into the 2.2.1 subsection, P6, L11-18. Given the hydrological setting of our study area, we assume that there is no significant vertical stratification so that the main driver for disconnection would be the water table drawdown due to permanent pumping.

2) Hydrogeology in an urbanised area: I was uncomfortable with some of the assumptions used in the generation of the water table map, in particular that there is no recharge in urban areas due to the presence of 'impervious' structures. This is not consistent with findings elsewhere, which have shown that 'impervious' structures like road networks are never completely so. Moreover, most large old cities have large old

and leaky water distribution and sewerage networks. Whether this is the case for the study area I do not know but evidence must be provided to satisfy that no recharge occurs there. In addition and of consideration for the generation of the water table map, sewerage networks can also act as drains and thus impart an upper limit for the position of the water table.

=> The second refers to the statement made P3 L5 that recharge into urban environment is null. We thank you to point out this mistake which is a remaining artifact resulting from former version of the study. We propose to remove the two sentences since the methodology does not require to identify or quantify recharge.

3) Hydrogeological interpretation: It is probably necessary to compare the assessment made using a water table map with a more hydrogeologically-based interpretation. For example, Lamontagne et al. (Hydrological Processes 28: 1561-1572) provide a methodology where assessments of connectivity for rivers can be made using pairs of surface and groundwater level measurements for different hydrogeological properties of riverbeds and aquifers. Even though the authors may not have all the information required (e.g. riverbed hydraulic conductivity) the Lamontagne et al. approach can still be used in a sensitivity analysis context. Indeed, a sensitivity analysis should also be applied to the water table approach as well if possible - what is the potential error in the map?

=> The third point refers to the lack of quantification of reliability of the method to determine SW-GW connection status, that could be carried out applying the Lamontagne et al. 2013 methodology. This is true that the method presented in the paper is not compared to other methodologies. Indeed, such a sensibility analysis could be helpful to characterize the reliability of the methodology, however this would require to set a priori knowledge about riverbed hydraulic conductivity in order to estimate the SW-GW flow rate. As we said into the main answer, such a study constitutes the next step, after characterizing one static state of water table, that would be the description of the hydrological functioning through the description of dynamics and SW-GW flow rate. We

argued this point in the corrected version, see section 3.6 P11/L5-9 in that purpose.

The empirical approach used by the authors to evaluate connectivity (Fig. 3) could be flawed based on hydrogeological principles. The lack of response of a water table to variations in surface water level could be due to a number of factors other than a river being disconnected. Indeed, even when disconnected, the specific recharge rate and flux below a river will increase at higher water levels (due to a greater head and wetted area), thus a response of the water table is still possible. On the other hand, a river could be connected but the alluvial aquifer have a very low transmissivity, which could result in a subdued variation in the water table especially if bores are at some distance from the river or the variation in river stage only for a short period of time. At a very low transmissivity, the response will only be notable when very close to the river. For a given change in river level, there will be a certain distance where the water table response can be practically measured considering other sources of variation.

=> Considering the infiltration fluxes in our area, they are stable even during flood due to the embankment of the river, which creates an almost stable wetted perimeter when no over-flooding is observed, as in our case.

=> We agree about the low transmissivity argument even though in that case the water fluxes are negligible. Being negligible, it means that the river and the aquifer are almost disconnected in that case. However, for the accuracy of our analysis, we consider a distance between piezometers and the river that is short enough to ensure water table response during flood.

Finally, several changes where operated in the text regarding the minor comments that are addressed:

P2, Line 1:'Embankments' should probably be 'levees' if they raise the water level at which water would spill into a floodplain

=> Replacing Embankments by levees

[Figure]

P4. Use of unsaturated zone thickness: I did not understand why this was better than using the water table elevation. It would also be preferable to use the term 'vadose zone' instead of 'unsaturated zone' considering capillary effects are of interest.

=> The main reason for using unsaturated zone depth instead of water table elevation is explained into introduction P2/L28-30:

"This methodology, that targets the unsaturated zone depth (UZD) instead of the hydraulic head, leads to lower values of the standard deviation of the estimation error for unconfined aquifer in non30 urbanized area (Kurtulus and Flipo, 2012; Mouhri et al., 2013; Rivest et al., 2008; Sagir and Kurtulusÿ, 2017)." Furthermore, we added this point P3/L32-P4/L2: "Unlike water table, UZD can be considered as a continuous stationary variable. The supposed stationarity of a variable makes it usable for ordinary kriging methodologies. In other cases, more complex non-stationary geostatistics should be applied, requiring hypothesis about the estimated variable."

P.20. Basis for the 10 m threshold. Some additional justification is required for using this threshold to identify disconnected conditions. Drawdown cones can be much deeper, especially when transmissivity in the aquifer is low.

=> P4-L23-29, adding explanation for the data selection step:

"The first category regroups all samples where the UZD value is affected by the pumping wells. The second category is composed by the other samples. Information about the locations of pumping wells is required to identify these samples. The observed minimal UZD of depreciated areas can be use as a threshold value to differentiate affected points from non-affected points. In this study, the samples with UZD value greater than 10 m are grouped in this category. Note that this value may vary according to the case study. This differentiation is required to elaborate a geostatistical tool (i.e. variogram model) that only depends of natural variability. Therefore, all the variographic studies are performed on this second category called unaffected UZD dataset."

P.6, line 11. This is only correct if the river level is constant. If the river level increases, the specific recharge rate and overall recharge flux below the river will increase because of a higher hydraulic head and a larger wetted area (i.e. wider river).

=> P6-L11-18, adding some consideration to be accounted for to describe infiltration rate during disconnection: "During the switching between connection status, the SW-GW connection status is considered as a transitional state, this condition can occur when the capillary zone intersects the riverbed (Brunner et al., 2009). The disconnected SW-GW condition can occur under different settings such as in case of high hydraulic conductivity contrast between the clogging layer and the aquifer (Brunner et al., 2009; Peterson and Wilson, 1988), the lowering of the water table (Dillon and Liggett, 1983; Fox and Durnford, 2003; Osman and Bruen, 2002; Rivière et al., 2014;Wang et al., 2011)) or the biological clogging of the riverbed (Newcomer et al., 2016; Xian et al., 2019). Considering a constant river water level and river width, the disconnection occurs when any further increase of the hydraulic head difference between the water table and the river water level does not affect the infiltration rate from the stream to the underlying aquifer, which remains constant."

Figures: Could in general all be improved - very faint lines and symbols in particular. Figure 6a was interesting and should be complemented with similar 2D cross-sections when feasible to get an idea of the shape of the water table near the river.

=> Legend and symbols were reworked for all figures.

The following illustration (Fig.1 of this answer) is an example of 2D cross sections of water table, river water level and topography. This figure shows the estimation of water table for LWC and HWC at connected and disconnected parts of the river. We think that this figure would bring redundant information with Fig.6 if included in the paper, this is why we would prefer not to do so.

Please also note the supplement to this comment:

https://www.hydrol-earth-syst-sci-discuss.net/hess-2019-101/hess-2019-101-AC2-supplement.pdf
* * *
[Figure]

[Figure]

**Fig. 1.** 2D cross-sections for profile 1 and profile 2. Profile 1 is always connected and profile 2 is always disconnected

**Supplement:**

[revised manuscript text omitted]

---

## Referee Comment (RC3) · Anonymous Referee #1 · 24 Jul 2019

Thank you for your reply to my comments. Nevertheless, I would like to point out a number of issues that have not properly been addressed. Below, page and line numbers refer to the new version of the article.

Regarding question (i) stated at the end of the introduction: I understand you do not want to make the suggested additional work, because you simply modified this question. It would have been more honest saying it explicitly... And it is a pity: the paper would greatly benefit from highlighting the improvements induced by the proposed methodological refinements.

Regarding question (ii) stated at the end of the introduction: you explicitly replied that

you do not address this question in this paper. Fair enough, but then, the question should be removed!!

P1 L14: Refer to Brunner et al. (2009) for these three connectivity status.

P1 L20: The reference to Bresciani et al. (2016) should better be put at the end of the sentence together with the reference already there.

P2 L14: Bresciani et al. (2018) did not use fuzzy logic or neural network. They used the diffusion kernel interpolation method. Please correct this.

P2 L27: "Thereafter" should be removed and a new sentence should start.

P2 L27: Change "shallow groundwater" for perhaps "relatively humid climate".

P4 L7-8: I still do not understand the reason for smoothing the DEM. How are your UZD measurements taken? If they are taken from dipper, I would think that the exact topographic level must be used, and not a smoothed one.

P4 L 8-9: Again, clarify the search radius. "in agreement with. . ." is cryptic. What did you precisely do?

P4 L10-13: This is also unclear: what did you do about this bias?

P4 L18-29: This part is still very much unclear. First of all, there should be a conceptual discussion about the effect of pumping. Namely, it must be recognized that the effect of pumping is not only punctual. Hence, nowhere will the water table be really natural/unaffected. In this view, it is not even clear why any data should be removed. Secondly, you write that the location of pumping wells is required. Do you have these data? It seems not, otherwise you would not need to use a threshold value on UZD. But this should be said! Thirdly, a rationale for the employed 10 m threshold is still lacking. Fourthly, you write that the variographic studies are performed on the second category of data. Does this mean that the rest of the analysis uses all the data from both categories?

P6 L19-30: Regarding the riverbed thickness (i.e. clogging layer), you replied that you assume that there is no clogging layer. Then, this assumption should be explicitly mentioned in the text.

P10 L16: Here I guess you are referring to the transition case, and not the disconnection case. The way you wrote this is quite confusing.

P10 L30: Why would the optimal value be reached when the relative numbers of matched cross sections are equal?? This would only be true in the very special case where there are equal numbers of connected and disconnected cross sections.

Fig. 2: River names are still lacking.

Also note that my comment on "favor river infiltration towards the aquifer" in Section 3.7 was not addressed at all. Please address it.

---

## Author Comment (AC3) · 21 Aug 2019

We thank you for this additional comment which helped us to identify the remaining unclear points of this paper. Please find hereafter a new rebuttal with the last version of the manuscript (supplements pdf).

*I understand you do not want to make the suggested additional work, because you simply modified this question. It would have been more honest saying it explicitly: And*

[Figure]

*it is a pity: the paper would greatly benefit from highlighting the improvements induced by the proposed methodological refinements.*

We apologize for the misunderstanding raised by our first answer to this point. Indeed, we want to keep our paper as a technical note. The comparison of the different interpolation methodologies is already the main topic of studies that are cited in to bibliographic review of the introduction (cf Introduction P2 L7-9). This comparison of kriging with other estimators was carried out in Varouchakis et al. 2013, we also added references that supports the choice of kriging methodologies (Emadi and Baghernejad, 2014, Adhikary and Dash, 2017 and Ohmer et al., 2017). The choice of UZD as a variable instead of hydraulic head is justified further in introduction (cf introduction P2 L33-35).

*Regarding question (ii) stated at the end of the introduction: you explicitly replied that you do not address this question in this paper. Fair enough, but then, the question should be removed!!*

Thank to point this out, we want to clarify our response. You must be referring to the question iv of the introduction which is: "(iv) finally, what are the consequences of such methodological refinements on produced maps of water table linked to hydrological events?". We believe that the paper answers this issue with a sensitivity analysis of the disconnection criteria. We remind that this paper is a technical note aiming at fully describing the method, and not to analyze the system functioning. This is what we meant by our former response: Âń These aspects are not developed in this study. We believe that such a development should constitute further works of the application of this study that we want to keep focus on the mapping issue. Âż.
*P1 L14: Refer to Brunner et al. (2009) for these three connectivity status.*

P1 L14-15: Done. We also added Dillon  Liggett (1983) and Fox  Durnford (2003) where these three connectivity status are described.

*P1 L20: The reference to Bresciani et al. (2016) should better be put at the end of the sentence together with the reference already there.*

Done, P1 L21

*P2 L14: Bresciani et al. (2018) did not use fuzzy logic or neural network. They used the diffusion kernel interpolation method. Please correct this.*

We add this methodology to our review of mapping methodology. All these information are added P2 L18-22

*P2 L27: "Thereafter" should be removed and a new sentence should start.*

Done P2

*P2 L27: Change "shallow groundwater" for perhaps "relatively humid climate".*

P2 L33-35 These words were replaced by the quote from the Haitjema and Mitchell-Brucker 2005 paper.

*P4 L7-8: I still do not understand the reason for smoothing the DEM. How are your UZD measurements taken? If they are taken from dipper, I would think that the exact topographic level must be used, and not a smoothed one.*

The reason for using a smoothed DEM is that with a high resolution DEM, topography contains more noise than water table. To remove this artifact we did a smoothing of the high resolution 25x25m DEM. This methology was proposed by Mouhri et al. 2013.

*P4 L 8-9: Again, clarify the search radius. "in agreement with: : :" is cryptic. What did you precisely do?*

The search radius is defined regarding two conditions: i) the DEM has to be smoothed enough to remove its high-resolution noise and ii) the information of river water level must be conserved in the final product. We tested many radius in order to fit with the conditions and found out a value approximating the average stream width (200m). We have added this explanation P4 L15-18

*P4 L10-13: This is also unclear: what did you do about this bias?*

The bias is induced by the smoothing effect. It is the difference between the true wellhead elevation and the smoothed DEM elevation. Thank to your comment, we found out that there was a mistake in the text P4 L23 where "between DEM data" was replaced by between smoothed DEM data".

*P4 L18-29: This part is still very much unclear. First of all, there should be a conceptual discussion about the effect of pumping. Namely, it must be recognized that the effect of pumping is not only punctual. Hence, nowhere will the water table be really natural/unaffected. In this view, it is not even clear why any data should be removed. Secondly, you write that the location of pumping wells is required. Do you have these data? It seems not, otherwise you would not need to use a threshold value on UZD. But this should be said! Thirdly, a rationale for the employed 10 m threshold is still lacking. Fourthly, you write that the variographic studies are performed on the second category of data. Does this mean that the rest of the analysis uses all the data from both categories?*

Some justifications were added P5 L3-7 to explain the identification of affected piezometers. The Dupuit-Forchheimer assumption is not valid within the capture zone of a pumping well. Therefore, the head is not hydrostatic (Grubbs et al., 1993), the piezometric value and the topography are not correlated within its capture zone. We noted that there is no correlation between the piezometric value and the topography for the locations where UZD exceeds 10 m (Fig.a). Without the exact knowledge of pumping wells location, we assume that these samples are located within the pumping wells capture zone. Indeed, this differentiation was only used to construct the experimental

variograms to which the variogram model are fitted. The rest of the analysis uses all the data from both categories.

*P6 L19-30: Regarding the riverbed thickness (i.e. clogging layer), you replied that you assume that there is no clogging layer. Then, this assumption should be explicitly mentioned in the text.*

P7 L3: mentioning the absence of clogging layer and its justification.

*P10 L16: Here I guess you are referring to the transition case, and not the disconnection case. The way you wrote this is quite confusing.*

We agree that mentioning the transition case improves the comprehension, it was corrected P10 L24.

*P10 L30: Why would the optimal value be reached when the relative numbers of matched cross sections are equal?? This would only be true in the very special case where there are equal numbers of connected and disconnected cross sections.*

We believe that in either case, the choice of the objective function to determine the optimal disconnection criteria is arguable. We would advice our readers to choose the more appropriate one depending on the case study, this was added P11 L6-10. We choose to change the used objective function from absolute number of cross-sections validation to relative number of cross-sections validation because we considered that it

would make it less dependent of the cross-section number. Indeed, this change does not change our conclusion because in our case, the number of disconnected sections is the same as the number of connected sections.

*Fig. 2: River names are still lacking*

Adding the river names in Fig.2

*Also note that my comment on "favor river infiltration towards the aquifer" in Section 3.7 was not addressed at all. Please address it.*

We apologize for the omission of this last point. We understand your comment about the map showed in Fig.6.b where stream is gaining in many portions of river. This issue is caused by the representation of the riverline of Fig.6 that conceals the local water table. Furthermore, the river water level is not represented in Fig.6.

We propose to add the information about the SW-GW relationship in connected sections (gaining, loosing and both) of river in the final maps showed in Fig.6. As it is mentioned in the corrected text (P12 - L1-5), the SW-GW relationship is established analyzing the difference between river water level and water table at a 2 pixels (50 m) distance from the river. The obtained relationship is attributed to riverline representation as a color code in Fig.6. Thanks to this new representation, the discussion about the effect of hydrological event on water table is reinforced (P12 L6-13).

Please also note the supplement to this comment:
https://www.hydrol-earth-syst-sci-discuss.net/hess-2019-101/hess-2019-101-AC3-

supplement.pdf

[Figure]

**Fig. 1.** Fig. a: Water table level compared to topographic level at LWC sampling points. The red symbols correspond to the samples affected by the pumping wells, the black symbols correspond to the unaffected

**Supplement:**

[revised manuscript text omitted]

---

## Author Response (AR1)

Dear editor,

We thank you for your consideration about our study, and for the feedback on the manuscript. As you mention it, we realize that our response could have appeared inappropriate regarding the chosen words. Indeed, we believe that the work of the reviewers helped us to raise several issues and inconsistencies of the initial version. By our responses, we are hoping that the reviewers still agreed with the corrections made in the paper.

Please, find hereafter, the corrections and the corresponding line in the latexdiff version, highlighting the changes that were operated after your comments:

*As suggested by reviewer 2, please also mention the potential for leaky sewer and water supply lumbing networks to recharge groundwater and act as drains (in the case of sewers) that limit how shallow the water table can rise in some areas.*

P1.L22-P2.L3: Adding precisions on the drain and recharge effect of underground hydraulic networks .

*Gaurantee*

P2.L27: changed

*directional trends in hydraulic head gradients*

P2.L33: changed

*word missing*

P3.L4-L6: The citation was corrected with the exact sentence, associated with a short description of the symbols used by Haitjema 2005.

*Not clear. Do you mean "subtracted from"? Is this the step where you convert the UZD map into a head map?*

P4.L11-L12: The "subtract" verb is the appropriate word, and it is indeed the step where the UZD map is converted into a head map, we corrected the word and added this precision to make the text clearer.

*This is not true as a blanket statement. It may be true in cases where the water table and topography, including their gradients, are variable spatially. But inherently the water table, topography and UZD are directional. This leads to a type of non-stationarity in which the expected value and it's variance differ dramatically when looking perpendicular to the gradient versus looking parallel to the gradient. In the former, at the local scale, there is no change in the variable, and the variance is nil. In the latter, at the local scale, the spatial variance increases parabolically and never reaches a sill if the gradient is uniform. So, perhaps the quick way out of this conundrum is for you to assert that the directional gradients in UZD are much less pronounced or more irregular than the h gradients, making it more amenable to treatment with stationary geostatistics. In the sentence that follows, you should make it clear that if there are significant trends in the data, then geostatistica approaches that account for that can be applied, such as universal kriging (cite a geostats text such as Goovaerts).*

P4.L18-22: We understand that this statement was not true in every topographic context or scale. We changed it following your argumentation. The new sentence compares qualitatively the stationarity of UZD and piezometric head, asserting that the UZD is more stationary than piezometric head.

*Computed*

P4.L25: changed

*Radii*

P4.L31: changed

*Ing*

P5.L7: changed

*While the above procedure was used to roughly approximate which wells are affected by pumping, any future applications of the method outlined in this technical note should identify the wells impacted by pumping using actual data on pumping rates and locations*

P5.L25-L27: sentence added

*Please show either in figure 3, 6 or as an inset in this figure which reach of the river-aquifer system is being depicted in a.*

Fig.5: the location of the studied reach for the disconnection criteria adjustment was directly added using a minimap that have the same extension than Fig.3 and 6.

Finally, we also mentioned the reviewers' contribution, as well as yours into the acknowledgments section.

---

## Editor Decision (ED1)

[revised manuscript text omitted]

Number: 1 Author:    Subject: Highlight    Date: 9/17/19, 11:47:57 AM

Not clear. Do you mean "subtracted from"? Is this the step where you convert the UZD map into a head map?

Number: 2 Author:    Subject: Highlight    Date: 9/17/19, 11:59:14 AM

This is not true as a blanket statement. It may be true in cases where the water table and topography, including their gradients, are variable spatially. But inherently the water table, topography and UZD are directional. This leads to a type of non-stationarity in which the expected value and it's variance differ dramatically when looking perpendicular to the gradient versus looking parallel to the gradient. In the former, at the local scale, there is no change in the variable, and the variance is nil. In the latter, at the local scale, the spatial variance increases parabolically and never reaches a sill if the gradient is uniform. So, perhaps the quick way out of this conundrum is for you to assert that the directional gradients in UZD are much less pronounced or more irregular than the h gradients, making it more amenable to treatment with stationary geostatistics. In the sentence that follows, you should make it clear that if there are significant trends in the data, then geostatistica approaches that account for that can be applied, such as universal kriging (cite a geostats text such as Goovaerts).

Number: 3 Author:    Subject: Inserted Text         Date: 9/17/19, 12:00:11 PM

computed

Number: 4 Author:    Subject: Inserted Text         Date: 9/17/19, 12:01:13 PM

radii

Number: 5 Author:    Subject: Cross-Out    Date: 9/17/19, 12:01:42 PM

Number: 6 Author:    Subject: Cross-Out    Date: 9/17/19, 12:02:06 PM

Number: 7 Author:    Subject: Inserted Text         Date: 9/17/19, 12:02:13 PM

ing

**2.1.2 Hard data selection & variograms**

[revised manuscript text omitted]

Number: 1 Author: Subject: Highlight Date: 9/17/19, 12:29:19 PM

Please show either in figure 3, 6 or as an inset in this figure which reach of the river-aquifer system is being depicted in a.